# Start learning coding without computers? A case study on children's unplugged gamified coding education tool with explanatory sequential mixed method

**Miao Huang, Lei Wang**◯*

School of Animation and Digital Arts, Communication University of China, Nanjing, China

* wanglei@cucn.edu.cn

## Abstract

Instruction in coding for children has emerged as a significant means of fostering computational thinking, with gamification serving a crucial reinforcing function in this educational process. This experimental study integrates four principal gamification elements—role-playing, rewards, challenges, and cooperation—into unplugged children's coding education tools to examine their impacts on children's flow experience and learning engagement. Under the theoretical framework of Stimulus-Organism-Response (S-O-R), researchers developed an unplugged coding education prototype named "Coding Adventure," employing an explanatory sequential mixed-methods approach with 295 Chinese elementary students (aged 8–10 years). Subsequent qualitative interviews were conducted with 12 stratified participants. Empirical findings demonstrate that role-playing and cooperation effectively enhance children's flow experience when engaging in coding education tools. Children role-play heroes and engage in "real fights," engage in teamwork and communicate with others, which can bring them into a flow experience. The flow experience effectively enhances children's learning engagement. The main reasons are the immersion, upgrade experience, practicality, goal-orientedness, teamwork and partner's suggestions brought by gamification elements. Moreover, rewards and cooperation also directly positively influence children's learning engagement. Intrinsic and extrinsic rewards, engagement in teamwork and receiving encouragement and suggestions from interactions with classmates are thought to increase children's motivation to learn. This study examined the effects of diverse gamification elements on children's flow experience and learning engagement in an unplugged gamified coding education tool under the framework of the S-O-R theory. Additionally, this study demonstrated important practical implications by providing developers of coding education tools with a clear path to enhancing participants' sense of immersion and achievement.

**Data availability statement:** The data that supports the findings of this study is openly available in https://figshare.com/ at https://doi.org/10.6084/m9.figshare.24942342.v1.

**Funding:** This study was financially supported by the Jiangsu Provincial Educational Science Planning Project Key Project (2022), issued by the Jiangsu Provincial Education Science Planning Leading Group and Communication University of China, Nanjing in the form of a grant (B/2022/04/42) received by HM. This study was also financially supported by the Jiangsu Provincial Department of Education in the form of an award (Achievements of Young and Middle-aged Academic Leaders in Jiangsu Universities' "Qinglan Project"; 2025) received by WL.

**Competing interests:** The authors have declared that no competing interests exist.

## 1. Introduction

Computational thinking (CT) has become a new standard for global users to integrate into digital products and information services [1]. Within China's educational landscape marked by heightened societal emphasis on childhood development, pedagogical institutions are progressively institutionalizing CT cultivation through systematic integration of coding education in early childhood and elementary education systems [2]. This pedagogical innovation frequently employs gamified instructional architectures to simultaneously optimize cognitive engagement and foster CT development, strategically preparing young learners for emerging technological paradigms dominated by robotic and artificial intelligence advancements [2].

Gamified learning refers to educators adding game-specific design elements to non-game educational situations [3]. The implementation of gamification strategies has the potential to enhance students' engagement in the learning process while simultaneously bolstering their motivation to address and resolve problems [4]. Nevertheless, it has been observed that Chinese parents typically express reservations regarding computer-based instructional approaches for their children [5]. They contend that such methods may adversely impact children's visual health and pose challenges to their self-regulation [5]. Furthermore, children may encounter feelings of loneliness as a result of prolonged engagement with computer-based and online learning, which can subsequently contribute to mental health issues [6]. Consequently, unplugged coding education tools have the potential to effectively address the physical and mental health issues that children may experience through face-to-face interactions [7,8].

Unplugged physical coding education represents a developmental compromise for children; however, it has garnered significant acclaim from educators for its effectiveness in promoting coding literacy among younger demographics [9]. In contemporary educational settings, visual coding education tools such as Scratch and ScratchJr exhibit limited efficacy for children, particularly in instances where educators lack comprehensive coding proficiency [10]. Furthermore, flowblocks, as a prevalent model of unplugged coding education tools, can significantly enhance children's self-efficacy [11].

In typical educational settings, educators may contemplate organizing unplugged gaming activities to facilitate children's engagement with foundational concepts of CT [12]. In the realm of physical education, designers of educational tools can capitalize on children's inclinations towards cartoon imagery, narratives featuring heroes, props, and small animals. By incorporating these elements into educational games, they can enhance the learning experience [13]. These educational tools have the potential to be further refined into treasure hunting games, which would enable children to engage their natural curiosity for exploration while utilizing the classroom setting to simulate game scenarios [14]. From the perspective of educational outcomes, unplugged coding education has been shown to significantly enhance elementary students' CT and cognitive abilities [15]. Notably, when integrated with gamification, it can effectively increase learners' engagement and communication skills, thereby making the learning experience more interactive and enjoyable [16].

While prior studies have suggested a range of solutions for unplugged gamified coding education tools for children, there is a notable lack of instances where these tools have been incorporated into assessments. Furthermore, investigations into the impact of these tools on children's learning outcomes are exceedingly limited. Consequently, the research objectives were established to develop a prototype for unplugged coding education and to evaluate: 1) the effects of the incorporated gamification elements; 2) the influence on children's flow experience; and 3) the effect on children's learning engagement.

## 2. Literature review

### 2.1. Gamification elements (role-playing, reward, challenge, cooperation)

Gamification involves a variety of design elements, many of which frequently interact synergistically [17]. In the case of gamification applied to education, role-playing, reward, challenge, and cooperation have received the focus of researchers as gamification design elements [17–19]. Combining gamified role-playing with reward, challenge, and cooperation for education management can effectively stimulate elementary students' enthusiasm for learning [20].

Role-playing games can bring to the field of education fun, positive and social characteristics [21]. In the context of role-playing, instructional designers can incorporate diverse reward elements such as points, medals, and rankings, and can also add challenges and punishments as motivational elements [22]. Rewards and punishments (feedback), as important components of the gamified incentive system, effectively promote children's behavioural motivation [23]. Challenge-based and achievement-based gamified design focuses on encouraging users to overcome difficulties, make progress, and ultimately feel empowered [24]. The advantages of gamified cooperation in the education field encourage students to engage in a cooperative environment through role-playing mutual assistance, resource sharing, teamwork and even moderate competition [25].

### 2.2. Flow experience in gamified education

According to the flow theory proposed by Csikzentimihalyi [26], flow refers to "the holistic sensation that people feel when they act with total involvement." (pp. 36). Flow experience can be further understood as the best psychological state felt by an individual, an optimal experience that maintains an individual's state of excitement or enjoyment [27]. Flow experience is a psychological activity highly related to learning experience [28]. Gamification is considered one of the most effective ways to stimulate flow experiences [29]. Gamified education improves students' flow experience in classroom learning, thereby improving their learning results [30]. Educators or designers cannot provide students with flow, but they can provide students with gamification elements to help them enter the flow state [31].

### 2.3. Children's learning engagement

Learning engagement, conceptualized as an active and fulfilling cognitive state during instructional processes, has been extensively validated as a pivotal determinant of academic achievement across educational contexts [32,33]. Children's enthusiasm for learning in primary school has been found to predict their future academic achievement, work status and life happiness [34]. Within the Chinese sociocultural context, elementary students' learning engagement is particularly emphasized as a crucial mechanism for social mobility [35,36]. This cultural prioritization establishes learning engagement as a critical observational metric in Chinese primary education systems, with substantial implications for long-term developmental trajectory. In the platform construction of computer science education, educators use gamification technology to develop learning environments for students to attract students and make them actively engage in interesting, exciting, and engaging courses [37]. Moreover, developmental research emphasizes the critical role of teacher-student interactions in classroom ecosystems, particularly regarding younger learners' sustained engagement in educational activities [38].

## 3. Theoretical foundation and hypotheses development

The stimulus-organism-response (S-O-R) theory was founded in 1974 [39]. The original intention of creating the theory was to study the consumption behaviour of individuals [40,41]. Since then, the S-O-R theory has also been extended to the study of flow experience stimulated by gamification, proving that flow experience mediates the relationship between gamification and learning engagement [42]. On this foundation, the researchers applied the four gamification elements (role-playing, reward, challenge and cooperation) suitable for children's unplugged coding education tool design in the literature review to refine the S-O-R theory based on gamification flow research, and obtained the following theoretical framework. Fig 1 shows theoretical framework of this study.

The incorporation of gamification elements significantly improves the flow experience of players during gaming by establishing objectives that necessitate cognitive engagement [43]. Role-playing can provide users with imaginative immersion, allowing them to be fully immersed in the experience through role identification and imaginary cognition [44]. Game rewards that stimulate flow enhance sympathetic activity and reduce parasympathetic inhibition in the brain, which are necessary for effective gameplay [45]. Challenge is at the heart of player engagement; it provides obstacles that players must overcome and is the primary source of entertainment in games [46]. Cooperation is in line with the social instinct of human beings, which is to work together for the common good, and it will lead to a more positive player experience [47].

Role-playing, game plot and narrative constitute an immersive gaming experience that draws users into the game world and evokes a state of complete immersion in them [48]. In addition, role-playing provides children with a platform to express emotions, understand social roles, and develop empathy, which is essential to their overall emotional intelligence [49]. Therefore, the following hypothesis is proposed:

**H1: Role-playing positively influences children's flow experience in the coding education tool.**

Flow theory predicts that the balance between gamified task difficulty and individual operating skill will lead to high levels of intrinsic reward, ultimately motivating the allocation of cognitive control [50]. Such intrinsic rewards encourage individuals to focus entirely on the activity itself and gain a strong sense of inner happiness, thereby achieving the flow experience [51]. Therefore, the following hypothesis is proposed:

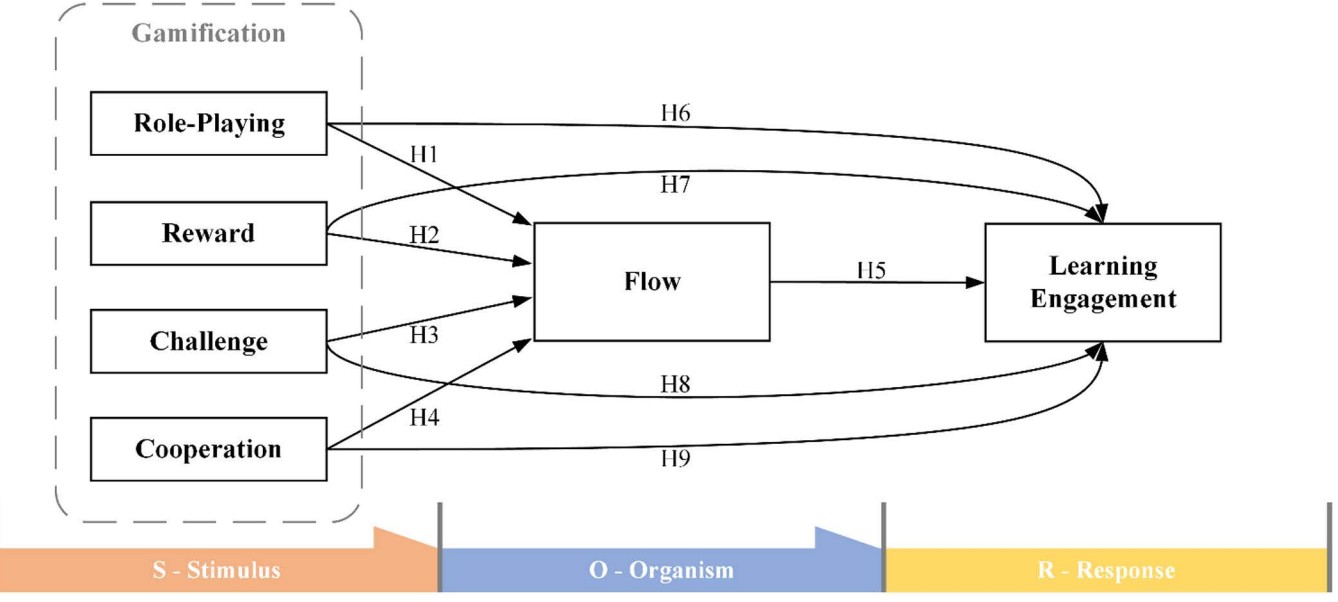

**Fig 1. Theoretical framework.**

**H2: Rewards positively influence children's flow experience in the coding education tool.**

Gamified challenges should match the user's ability level, provide clear goals and immediate feedback to promote the flow state within the game [52]. The study by Anunpattana et al. [53] used flow theory to explain the process of gamified challenges affecting students' abilities, in which they detected that as the challenge increases and abilities increase, students' flow experience will further enter the arousal zone. Therefore, the following hypothesis is proposed:

**H3: Challenges positively influence children's flow experience in the coding education tool.**

In multiplayer games, effective communication and teamwork, and sharing relevant information about game tasks with team members, effectively stimulate group flow in cooperative games [54]. Compared with the competitive virtual environment experience, the cooperative mode stimulates higher levels of emotion and decision-making challenges, and also promotes users' flow experience [55]. Therefore, the following hypothesis is proposed:

**H4: Cooperation positively influences children's flow experience in the coding education tool.**

Students experienced a flow state while applying learning games with virtual reality technology, which provided them with a high sense of immersion and presence, promoting their learning engagement [56]. Park [57] found that learning flow is a predictor of learning engagement and teaching presence in coding education. Therefore, the following hypothesis is proposed:

**H5: The flow experience positively influences children's learning engagement in the coding education tool.**

The quasi-experimental study by Yang et al. [58] obtained comparative results, proving that role-playing-based CT education strategies show a significant effect in promoting students' learning engagement and motivation. In addition, role-playing strategies have also been shown to enhance students' learning engagement and critical thinking in online teaching [59]. Therefore, the following hypothesis is proposed:

**H6: Role-playing positively influences children's learning engagement in the coding education tool.**

Students gain inner pleasure and satisfaction through learning activities and feel intrinsic rewards; they also feel extrinsic rewards by receiving praise and encouragement from society and family, as well as scholarships or career development given by others [60]. When children complete their learning progress, the gamification system quickly gives back rewards to provide a sense of achievement, thereby stimulating children's stronger desire to learn [61]. Therefore, the following hypothesis is proposed:

**H7: Rewards positively influence children's learning engagement in the coding education tool.**

Compared with traditional education methods, challenge-based gamified education has a more positive effect on students' learning [24]. The use of challenge-based gamified learning methods by students effectively improved their academic performance levels and overall motivation [62]. Therefore, the following hypothesis is proposed:

**H8: Challenges positively influence children's learning engagement in the coding education tool.**

Empirical studies demonstrate that children engaged in cooperative learning, particularly during coding education, achieve superior learning outcomes compared to those learning individually [63]. For instance, students with lower academic performance show significant learning gains through structured cooperation with higher-achieving peers [63]. Moreover, effective collaboration in serious games is facilitated by the cognitive management of diverse perspectives, the adaptation to the strengths and weaknesses of the group, and the pursuit of shared objectives during gameplay, all of which can enhance the learning experience [64]. Therefore, the following hypothesis is proposed:

**H9: Cooperation positively influences children's learning engagement in the coding education tool.**

## 4. Methodology

### 4.1. Research design

Given that the participants in this study are still in their developmental stages and possess limited comprehension and expressive capabilities, the findings derived from a singular research methodology may not accurately reflect the broader population. Consequently, this research employed an explanatory sequential mixed methods approach to mitigate potential biases associated with utilizing a singular methodology. By integrating both quantitative and qualitative techniques, the study aimed to elucidate the underlying factors contributing to the observed phenomenon [65]. In this study, the explanatory sequential mixed methods approach was implemented in two distinct stages: a quantitative survey followed by qualitative interviews [66].

In addition, the study has been assigned study protocol code 2023/CUCN/IRB/0009, which should be used for all communications related to this study. The recruitment process for this study commenced on March 3, 2023, and is scheduled to conclude on March 2, 2024. The research protocol secured threefold consent through participants' verbal assent, institutional approval from school administrators and homeroom teachers, and written guardian authorization. In the first stage, quantitative data was collected to ascertain preliminary findings, which were subsequently elaborated upon through a detailed analysis of the quantitative results [67]. Due to the need to observe children's feedback on specific unplugged coding education tools, the researchers designed a prototype for coding education that meets the research objectives in the preparation phase. Fig 2 shows the detailed steps of this study.

### 4.2. Prototype design and application

To incorporate four gamification elements (role-playing, reward, challenge and cooperation) into the prototype of this study, the researchers initially sought established unplugged coding education tools available in the market as a reference for enhancement. While all identified tools exhibit varying levels of playfulness, none were found to encompass all four gamification elements, and several are also subject to copyright restrictions. Therefore, the researchers designed a prototype that fully meets the experimental requirements, called "Coding Adventure", which integrates the established gamification elements analysed in the study through the following design content (see Table 1).

"Coding Adventure" is a maze-based game prototype structured around a grid of tiles. Players assume the role of valiant knights embarking on a quest that encompasses a variety of scenes across multiple levels. The complexity of the maps escalates in tandem with the increasing difficulty of the game. Prior to commencing their journey, players are afforded the opportunity to examine the map and strategize their route. Subsequently, they select the appropriate coding command cards that correspond to their planned route and the obstacles they may encounter. As players advance through the game, they utilize the command cards at their disposal to execute movements and engage in interactive actions. Successfully navigating the maze or vanquishing adversaries is regarded as a triumph, whereas the premature

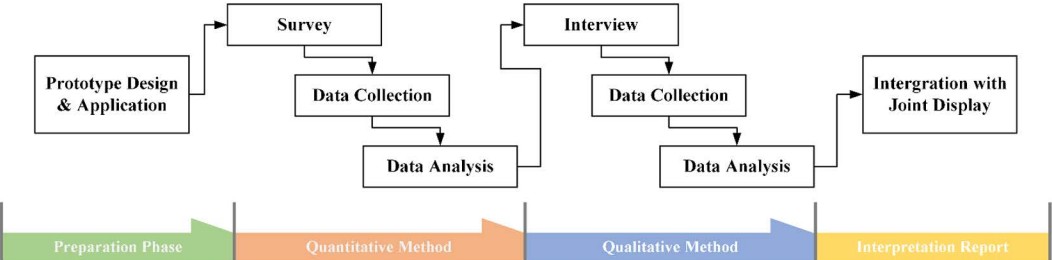

**Fig 2. The research process based on the explanatory sequential mixed method (Creswell, 2014).**

**Table 1. Design of "coding adventure" prototype.**

| Gamification Element | Specific Game Rules |
|---|---|
| Role-playing | 1. Play the role of knights |
| | 2. Wear safe props as equipment |
| | 3. Be told the adventurous game story before the game starts |
| | 4. Exploration and rescue as mission goals |
| Reward | 1. Get small dolls as rewards after passing levels |
| | 2. Once defeat monsters, the level and skills of characters will be upgraded |
| | 3. Completing ordinary tasks will get one doll, and completing difficult tasks will get double the number of dolls |
| Challenge | 1. Each level has multiple routes |
| | 2. Monsters and roadblocks as obstacles to pass the level |
| | 3. Difficulty will increase as the game progresses |
| | 4. Monsters have special counterattacks |
| Cooperation | 1. Allows multiple players at the same time |
| | 2. Children can plan with their teammates before the game starts |
| | 3. Children can ask for help from classmates outside the game when they encounter difficulties |
| | 4. Classmates can watch and encourage |
| | 5. Teachers will provide information tips |

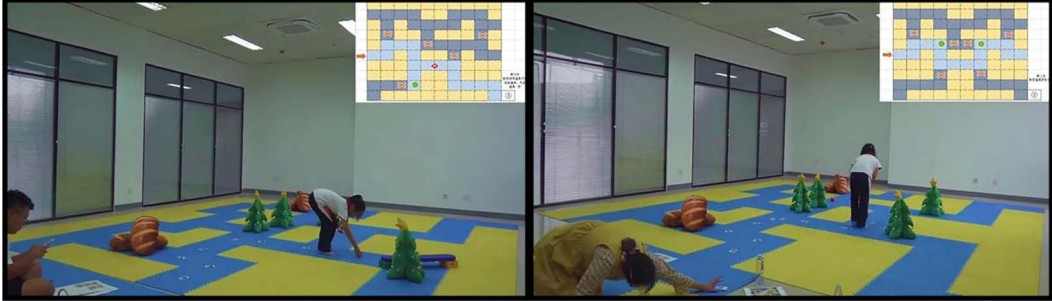

**Fig 3. Testing of prototype "coding adventure" (Source: Authors' work).**

depletion of all command cards is classified as a failure. The game scenarios are constructed exclusively using readily available and safe props within classroom settings, thereby facilitating the involvement of multiple participants. The testing of the prototype is shown in Fig 3.

After the prototype design of "Coding Adventure" was completed, the researchers applied to enter twelve classes in two primary schools for testing in Nanjing, China. The entire testing process was approved by the primary schools and conducted openly under the supervision of teachers in the schools. The researchers used positive guidance and kind language throughout the test to ensure that all children received positive evaluations and avoided any situations that might harm them.

During the test, the researchers uniformly described the gamification features of "Coding Adventure" to children, weakening the concepts of "education" and "tool" to enhance their interest in engagement. Therefore, the "game" expressed by children below refers to this prototype. After the test was completed, the researchers distributed questionnaires to all children who completed the game and collected them. Finally, 312 questionnaires were obtained, of which 295 were complete and met the requirements for data analysis. In the subsequent qualitative interview phase, the researchers randomly selected twelve children who completed the questionnaire as interviewees.

### 4.3. Data collection and analysis

**4.3.1. Quantitative survey.** In the post-test quantitative survey, the researchers designed survey items based on previous studies and used the cartoon shapes in the prototype as the design background to create a colourful paper questionnaire. The researchers distributed questionnaires to the children to answer after they completed the prototype experience. The children who completed the questionnaire carefully received cartoon stickers as small rewards. The questionnaire items measuring gamification were methodologically adapted from Grangeia et al.'s [68] scale, items assessing flow were empirically modelled after Thomas and Baral's [42] framework, and items evaluating learning engagement were systematically constructed based on Yang et al.'s [69] validated instrument (Appendix A in S1 File). In order to conform to children's understanding and response habits, the researchers have simplified and adjusted the expression of items. English items were translated into Chinese during the survey and verified by professional translators. The items were evaluated using a five-point Likert scale (1 for strongly disagree, 2 for disagree, 3 for neutral, 4 for agree, and 5 for strongly agree) [70].

The sample of study was a total of 295 participants. Since the children engaging in this test are concentrated in the second to fourth grades of primary school, the age range of the survey participants is 8–10 years old. The study included a marginally higher number of male participants (51.19%) compared to female participants (48.81%); however, the overall gender ratio remained balanced. Most of them have no experience in coding education (67.12%), and most of the rest have only limited experience (25.76%). See the Table 2 below for specific data.

**4.3.2. Qualitative interview.** After completing the quantitative survey, the researchers designed interview questions based on the results of the analysis. In order to obtain more feedback from children, the interview questions focused on children's views on gamification design elements (role-playing, reward, challenge and cooperation) during the prototype testing process, avoiding the "flow" and "learning engagement" themes, which are difficult for children to understand. The interview questions are summarized in Appendix B in S1 File.

The interview participants were randomly selected by researchers from the children who completed the questionnaire—a total of twelve children. The researchers conducted face-to-face interviews with each of them for about 15 minutes. Participants were coded according to standards for systematic interview analysis and to protect their privacy [71]. See the Table 3 below for specific information.

## 5. Results

Following the research process of the explanatory sequential mixed method, this study successively displays the results of quantitative surveys and qualitative interviews, and finally integrates and displays them to achieve the effect of explanation. Quantitative data was analysed with partial least squares structural equation modelling (PLS-SEM) using Smart-PLS software (Version 4.0), while qualitative data was analysed with thematic analysis (TA) using MAXQDA software (Version 2022).

**Table 2. Survey demographic profile.**

| Category | Option | Total N = 295 | |
|---|---|---|---|
| | | **Frequency** | **Percentage** |
| Age | 8 | 63 | 21.36% |
| | 9 | 91 | 30.85% |
| | 10 | 141 | 47.80% |
| Gender | Male | 151 | 51.19% |
| | Female | 144 | 48.81% |
| Coding learning experience | Never | 198 | 67.12% |
| | Less than one year | 76 | 25.76% |
| | One year and above | 21 | 7.12% |

**Table 3. Interview demographic profile.**

| Code | Gender | Age | Coding Learning Experience |
|------|--------|-----|----------------------------|
| IK01 | Female | 10 | Less than one year |
| IK02 | Male | 8 | Never |
| IK03 | Male | 9 | Never |
| IK04 | Female | 9 | One year and above |
| IK05 | Male | 10 | Never |
| IK06 | Male | 10 | Less than one year |
| IK07 | Female | 8 | Never |
| IK08 | Male | 10 | Never |
| IK09 | Female | 9 | Less than one year |
| IK10 | Male | 10 | Never |
| IK11 | Female | 10 | Never |
| IK12 | Female | 10 | One year and above |

### 5.1. Survey results

**5.1.1. Measurement model.** Table 4 focuses on factor loadings, Cronbach's alpha (CA), composite reliability (CR) and average variance extracted (AVE) in the measurement model. The fact loading of all items is greater than 0.708, the CA of all variables remains between 0.7–0.95, and the AVE exceeds 0.5, which shows the validity of items and variables [72].

The researchers applied two criteria to detect the discriminant validity of variables, namely the Fornell-Larcker criterion and the Heterotrait-Monotrait (HTMT) criterion. Table 5 shows the discriminant validity of the latent variables in this study using the Fornell-Larcker criterion, which requires that the square root of the AVE of the latent variable is greater than the correlation coefficient between the latent variable and all other latent variables [73,74]. Table 6 demonstrates HTMT discriminant validity across measured variables, as evidenced by inter-construct correlation coefficients ranging from 0.29 to 0.795, all falling below the stringent threshold of 0.85 recommended for behavioural research [73,74]. Through the data verification of Tables 5 and 6, there is no overlap or unclear distinction between the variables in this study.

**5.1.2. PLS-SEM structural model.** In the PLS-SEM structural model of this study, H1, H4, H5, H7, and H9 are proven to have significant positive effects because their *p*-values are less than 0.05 and their *β* values are positive [72]. Moreover, the *t*-values of these established hypotheses are all greater than 1.96, which indicates that they are significant at the 95% confidence level [75]. Therefore, the quantitative results show that role-playing and cooperation in gamification elements have significant positive effects on children's flow experience, and flow experience has a significant positive effect on children's learning engagement. In addition, rewards and cooperation have been proven to have significant positive effects on children's learning engagement (Table 7).

The researchers also examined the value of $f^2$ to determine the effect size of established hypotheses [72]. This study applied the measurement standards for effect size ($f^2$) proposed by Kenny [76]: large (0.025), medium (0.01) and small (0.005). According to the above standards, excluding the influence of reward on children's flow experience, which can be judged as medium effect size, the influence of role-playing and cooperation on flow, the influence of flow on learning engagement, and cooperation on learning engagement can all be judged as large effect size. See the Table 8 below for details.

### 5.2. Interview results

In this qualitative study, the researchers classified and linked the data collected during the interviews according to the themes of this study, and concentrated the representative quotes [71]. Given the cognitive levels of children, the qualitative interviews focused solely on gamification elements without addressing targeted inquiries into their flow experience or learning engagement. Nonetheless, the researchers conducted a more in-depth investigation into the children's flow

**Table 4. The Measurement Model Assessment Results.**

| Construct | Item | Loading | CA | CR (rho_a) | CR (rho_c) | AVE |
|---|---|---|---|---|---|---|
| Role-playing (RP) | RP1 | 0.918 | 0.940 | 0.940 | 0.957 | 0.848 |
| | RP2 | 0.936 | | | | |
| | RP3 | 0.917 | | | | |
| | RP4 | 0.912 | | | | |
| Reward (RE) | RE1 | 0.830 | 0.831 | 0.836 | 0.887 | 0.663 |
| | RE2 | 0.817 | | | | |
| | RE3 | 0.790 | | | | |
| | RE4 | 0.819 | | | | |
| Challenge (CH) | CH1 | 0.848 | 0.918 | 0.919 | 0.942 | 0.803 |
| | CH2 | 0.932 | | | | |
| | CH3 | 0.923 | | | | |
| | CH4 | 0.880 | | | | |
| Cooperation (CO) | CO1 | 0.892 | 0.926 | 0.930 | 0.947 | 0.818 |
| | CO2 | 0.935 | | | | |
| | CO3 | 0.928 | | | | |
| | CO4 | 0.860 | | | | |
| Flow (FL) | FL1 | 0.892 | 0.898 | 0.913 | 0.925 | 0.714 |
| | FL2 | 0.905 | | | | |
| | FL3 | 0.897 | | | | |
| | FL4 | 0.712 | | | | |
| | FL5 | 0.804 | | | | |
| Learning engagement (LE) | LE1 | 0.944 | 0.940 | 0.952 | 0.957 | 0.848 |
| | LE2 | 0.950 | | | | |
| | LE3 | 0.944 | | | | |
| | LE4 | 0.841 | | | | |

CA = Cronbach's alpha; CR = Composite reliability; AVE = Average variance extracted

**Table 5. Discriminant validity (Fornell-Larcker criterion).**

| | RP | RE | CH | CO | FL | LE |
|---|---|---|---|---|---|---|
| **RP** | 0.921 | | | | | |
| **RE** | 0.658 | 0.814 | | | | |
| **CH** | 0.457 | 0.273 | 0.896 | | | |
| **CO** | 0.464 | 0.256 | 0.733 | 0.904 | | |
| **FL** | 0.493 | 0.372 | 0.404 | 0.464 | 0.845 | |
| **LE** | 0.511 | 0.406 | 0.429 | 0.563 | 0.646 | 0.921 |

experience by inquiring about their play processes by follow-up questions during the interviews. Additionally, the researchers gained further insights into the children's learning engagement by soliciting information regarding their learning outcomes. This data has been coded in the MAXQDA software according to commonly used keywords by the respondents, facilitating the categorization of responses into corresponding themes (Table 9). These codes, along with quantitative results, are integrated in the subsequent section to fulfil the requirements of an explanatory sequential mixed methods research approach.

**Table 6. Discriminant validity (HTMT criterion).**

|    | RP | RE | CH | CO | FL | LE |
|----|-----|-----|-----|-----|-----|-----|
| **RP** | | | | | | |
| **RE** | 0.743 | | | | | |
| **CH** | 0.490 | 0.309 | | | | |
| **CO** | 0.498 | 0.290 | 0.795 | | | |
| **FL** | 0.531 | 0.413 | 0.445 | 0.505 | | |
| **LE** | 0.537 | 0.452 | 0.457 | 0.600 | 0.695 | |

**Table 7. Hypotheses testing results.**

| Hypothesis | | β-value | Std. dev | t-value | p-value | Discussion |
|----|----|----|----|----|----|----|
| H1 | RP ->FL | 0.265 | 0.080 | 3.393 | 0.001 | Accept |
| H2 | RE ->FL | 0.114 | 0.066 | 1.677 | 0.094 | Reject |
| H3 | CH ->FL | 0.056 | 0.075 | 0.659 | 0.510 | Reject |
| H4 | CO ->FL | 0.274 | 0.082 | 3.327 | 0.001 | Accept |
| H5 | FL ->LE | 0.426 | 0.068 | 6.199 | 0.000 | Accept |
| H6 | RP ->LE | 0.100 | 0.077 | 1.311 | 0.190 | Reject |
| H7 | RE ->LE | 0.114 | 0.057 | 1.997 | 0.046 | Accept |
| H8 | CH ->LE | −0.072 | 0.083 | 0.824 | 0.410 | Reject |
| H9 | CO ->LE | 0.343 | 0.092 | 3.712 | 0.000 | Accept |

**Table 8. The effect sizes of hypotheses.**

| Hypothesis | | β-value | p-value | f² | Effect Size |
|----|----|----|----|----|----|
| H1 | RP ->FL | 0.265 | 0.001 | 0.050 | Large |
| H4 | CO ->FL | 0.274 | 0.001 | 0.049 | Large |
| H5 | FL ->LE | 0.426 | 0.000 | 0.260 | Large |
| H7 | RE ->LE | 0.114 | 0.046 | 0.015 | Medium |
| H9 | CO ->LE | 0.343 | 0.000 | 0.104 | Large |

## 5.3. Results integration

This study conducts joint display through the pillar integration process (PIP) [77]. The results of this study are placed in the middle of the table as "pillar building themes", with quantitative and qualitative data presented on either side of it for comparison and presentation [77] (see Table 10).

The interviews for the qualitative part were arranged in a short time after the children participated in the game, so as to restore the children's intuitive feelings as much as possible. This arrangement also brought problems, which is that the excitement brought by the game made the children's feedback mostly positive. These qualitative feedbacks facilitated the researchers to explain the positive quantitative results, but also brought difficulties to the negative quantitative interpretation. Nevertheless, these feedbacks still contain a certain number of possible reasons for the falsified hypotheses, especially the reason why the role of the "challenge" gamification element was denied in all hypotheses. Therefore, the researchers also specially sorted out the interviewees' views on the failure of "challenge" in the prototype (see Table 11), which will be discussed below.

**Table 9. Interview themes and codes.**

| Topic/ Theme | Keyword/ Code |
|---|---|
| Role-Playing | Heroic fantasy |
| | Real fight |
| | Easily remember |
| | Immersion |
| | Feeling upgrade |
| Reward | Practicality |
| | Game goal |
| | Level award |
| | Physical prize |
| Challenge | Low difficulty setting |
| | Encourage cooperation rather than confrontation |
| Cooperation | Teamwork |
| | Communication |
| | Encouragement |
| | Good suggestion |
| | Encouragement |

## 6. Discussion

Drawing upon the integrated analysis of quantitative and qualitative findings, three principal insights emerge from this study. First, role-playing and cooperation in gamification elements demonstrate notable efficacy in enhancing flow experience within coding education tools. Second, the flow experience effectively promotes children's learning engagement. Third, rewards and cooperation also directly positively influence children's learning engagement. In the following, separate discussions are described in more detail based on the specific themes presented above.

The first theme highlights the importance of role-playing in play experiences. Specifically, role-playing shows a positive and efficient effect on children's flow experience in the learning process. The main reason is that role-playing allows children to experience the feeling of being heroes engaging in "real fights." Respondents felt that "playing as a knight" (IK07) "actually holding a weapon in the game" increased the realism of role-playing (IK05). In this process, children more clearly remember the operations performed and information obtained during the engagement process, and are immersed in the gamified environment, even temporarily forgetting the limitations of space and time. The substantial effect size observed in the quantitative findings further underscores the significant impact of role-playing. This evidence supports the assertion that role-playing can effectively enhance children's imagination, engage them in the gaming environment, and facilitate an immersive gaming experience [44,48].

The second theme summarizes the importance of cooperation in play experience. Specifically, cooperation positively and efficiently influences children's flow experience in the learning process. Respondents pointed out that "discussions with classmates during the game" (IK02) provided them with opportunities for communication (IK09) and encouragement (IK03). The main reason is that cooperation provides children with opportunities to engage in teamwork and communicate with their classmates. In addition, a good cooperation mechanism allows children to receive encouragement and suggestions from others once they encounter difficulties in practice. In the quantitative results on effect size, teamwork and communication in multiplayer cooperative games have been proven to be highly effective in stimulating group flow among gamers [62], thereby stimulating higher levels of emotion and decision-making challenges [55].

In the third theme, the importance of flow experience is focused. Specifically, children's flow experience of the coding education tool positively and efficiently influences their learning engagement. "Being immersed in the game" (IK04) and

**Table 10. Pillar integration of quantitative and qualitative data.**

| QUAN Data | | Pillar Building Themes | QUAL Data | |
|---|---|---|---|---|
| $\beta=0.265$, $t=3.393$, $p=0.001$ $f^2=0.050$ | Significant positive effect at the 95% confidence level Effect size is large | H1. RP ->FL (Accepted) | Heroic fantasy | "Playing as a knight in the game makes me feel like a hero saving the world." (IK07) |
| | | | Real fight | "It was impossible for me to actually hold a weapon in the games I played before, but your game makes me feel very real." (IK05) |
| | | | Easily remember | "After playing it in a real environment, I can easily remember the process of passing the level." (IK01) |
| | | | Immersion | "Playing such a large game on the playground made me forget about the outside world, as if I was really in the game." (IK03) |
| $\beta=0.274$, $t=3.327$, $p=0.001$ $f^2=0.049$ | Significant positive effect at the 95% confidence level Effect size is large | H4. CO ->FL (Accepted) | Teamwork | "If I have any questions, I feel reassured that I can discuss them with my classmates, so I can think about how to pass the test with peace of mind." (IK02) |
| | | | Communica-tion | "In the past, coding games were played by one person on the computer, and it was boring without being able to chat with classmates. I hope my friends can see what I do." (IK09) |
| | | | Encourage-ment | "My classmates cheered me on so that I could happily concentrate on the game." (IK03) |
| $\beta=0.426$, $t=6.199$, $p=0.000$ $f^2=0.260$ | Significant positive effect at the 95% confidence level Effect size is large | H5. FL ->LE (Accepted) | Immersion | "Being immersed in the game makes me feel that time passes very quickly, and I feel that I have learned a lot of coding skills." (IK04) |
| | | | Feeling upgrade | "Through character upgrading in the game, I can feel the role of coding. I must plan my growth route in order to better eliminate enemies." (IK06) |
| | | | Practicality | "I can feel that the skills in the game are useful. After I learn to use these skills, I can actually use these skills in coding software." (IK04) |
| | | | Game goal | "The setting of game goals allowed me to understand the process of learning coding from simple to difficult. I can now understand how complex games are made." (IK07) |
| | | | Teamwork | "It feels good to play games with my classmates. I also hope that I can often cooperate with my classmates to complete something while studying." (IK10) |
| | | | Good suggestion | "During the game, I learned that coding can also be a matter of brainstorming, and my classmates gave me a lot of good ideas. I think I can use this method in my study." (IK01) |
| $\beta=0.114$, $t=1.997$, $p=0.046$ $f^2=0.015$ | Significant positive effect at the 95% confidence level Effect size is Medium | H7. RE ->LE (Accepted) | Level award | "I really like games where you can get rewards for learning. I will keep upgrading and learning." (IK03) |
| | | | Practicality | "The skills I got after every level in the game are very useful and I think I can use them in Scratch on my pad." (IK12) |
| | | | Physical prize | "If I get good grades, my family will give me gifts. In order to get the gifts, I will study hard." (IK02) |
| $\beta=0.343$, $t=3.712$, $p=0.000$ $f^2=0.104$ | Significant positive effect at the 95% confidence level Effect size is large | H9. CO ->LE (Accepted) | Teamwork | "Through cooperation with classmates, I realize the power of teamwork, and I will apply this to my studies." (IK04) |
| | | | Good suggestion | "But when I encounter difficulties in the game, my classmates will give me suggestions, which I find very useful. I will also ask my classmates for their suggestions when I study in the future." (IK06) |
| | | | Communica-tion | "It's so fun to play games and chat in class. I think this is good, as it allows me to quickly learn coding knowledge." (IK09) |
| | | | Encourage-ment | "When I encountered difficulties, my classmates cheered me on and encouraged me so that I could try new solutions again." (IK07) |

**Table 11. Pillar integration of invalid gamification element in prototype.**

| QUAN Data | | Pillar Building Themes | QUAL Data | |
|---|---|---|---|---|
| $\beta=0.056$, $t=0.659$, $p=0.510$ | H3. CH ->FL (Rejected) | Challenge has no effect in this prototype. | Low difficulty setting | "This game is not difficult. I passed it in three attempts. I think you can always pass it if you keep trying. The monsters in the later levels look powerful, but they are actually easy to defeat." (IK02) |
| $\beta=-0.072$, $t=0.824$, $p=0.410$ | H8. CH ->LE (Rejected) | | Encourage cooperation rather than confrontation | "I feel that this game requires us to work together with our classmates to pass the levels, rather than competing with each other to see who is better, so I don't have to think about competing with my classmates." (IK05) |

"feeling character upgrading" (IK06) drove respondents' application of practical skills (IK04), setting of learning goals (IK07) and building of teamwork (IK01; IK10). The main reasons are the immersion, upgrade experience, practicality, goal-orientedness, teamwork and partner's suggestions brought by the gamification elements. In the flow experience, children gain a high sense of immersion and presence [56], which enhances their intrinsic interest and curiosity, thus enhancing their learning motivation [78]. Moreover, the results of the quantitative part prove that the effect of such flow experience is positive and large.

Through the fourth theme, the positive effects of rewards are summarized. Specifically, rewards positively promote children's learning engagement with coding education tools, but the effect is medium. "Getting rewards through learning" enables the participant (IK03) to "keep upgrading and learning." Children believe that the virtual rewards brought by upgrades and the practicality and physical rewards brought by skill improvement effectively strengthen their enthusiasm for learning (IK02; IK12). The inner pleasure that children gain from improving their learning activities is regarded as intrinsic rewards, while promotion from the outside world and direct financial rewards are regarded as extrinsic rewards, both of which can be key elements in influencing learning [60].

The fifth theme also adds the positive role of cooperation in learning engagement. Specifically, cooperation positively promotes children's learning engagement with coding education tools, and the effect is large. "Playing games and chatting in class" (IK09) brought "teamwork," (IK04) "helpful suggestion," (IK06) and "encouragement" (IK07) to children, which allowed them to "quickly learn coding knowledge" (IK09). Engaging in teamwork and receiving encouragement and suggestions from partners are seen by children as improving their motivation to learn. Engaging in learning activities together can increase the social intimacy between learners, thereby influencing neural synchronisation between each other and generating learning consensus [79].

The researchers also explored the reasons why "challenge" as a gamification element failed in this prototype through an additional theme. When designing the initial prototype, "challenge" was considered to be a gamification element with motivational functions. However, the quantitative analysis results showed that "challenge" did not show significant effects in all hypotheses, and the respondents ultimately believed that the difficulty setting of this prototype was "easy to defeat" (IK02), "competition with classmates was not encouraged" (IK05), and "there was a lack of clear victory goals" (IK07), which made the "challenge" element fail to play its due role in the participants' game process.

Feedback from respondents suggests that the ineffectiveness of "challenges" in this study may be attributed to variations in participants' backgrounds and their prior knowledge [80]. To facilitate successful task completion for all participating children, the prototype was designed with a low level of difficulty. However, this uniform approach to low-difficulty challenges may inadvertently exclude certain learners [81]. Furthermore, discrepancies between the design of the challenges and the course content could result in a disconnection between the gamified elements and the intended learning outcomes [82]. Future developers can re-evaluate the design of "challenge" related functions, correct the difficulty and settings of "challenge", and improve the user experience of unplugged children's coding education.

In addition, the results of the qualitative part also demonstrate the intrinsic connections of gamification elements in this prototype. Role-playing as a hero can often bring a sense of immersion to participants; teamwork can bring communication, suggestions and encouragement; and tangible rewards can bring more practical motivation. These gamification elements usually do not work independently, but are often mentioned in combination by participants. Related research also indicates that the combination of gamified sub-elements effectively enhances learners' motivation and engagement in learning [83].

## 7. Conclusion

### 7.1. Theoretical significance

The S-O-R theory has received considerable interest in the examination of how gamification components enhance learners' flow experiences [42]. This study investigated the relevance of the specified framework in relation to children's learning engagement within the framework of a gamified unplugged coding educational tool. An explanatory sequential mixed methods approach was employed to elucidate the underlying factors contributing to the observed phenomenon. In addition, the effects of role-playing and cooperation as stimulus (S) processes under the framework of the S-O-R theory are clearly evident. In summary, this study verified the applicability of the S-O-R theory in gamification to promote children's coding learning engagement.

### 7.2. Practical significance

The prototype titled "Coding Adventure," created for the purposes of this study, has undergone testing and has demonstrated effective functionality. During the data collection process, the unplugged gamified coding education tool, developed according to the researchers' design concept, garnered favourable assessments from children, thereby demonstrating its practical significance. The prototype underwent numerous iterations of level runs in an unplugged setting and successfully completed field testing, effectively engaging the children involved. The three gamification elements of role-playing, cooperation and rewards have been found to enhance children's learning engagement during prototype testing. Therefore, educators may consider enhancing children's sense of immersion and achievement by adding safe wearable devices, increasing physical rewards, and allowing on-site communication, while integrating knowledge points into games in the context of unplugged coding education. The importance of flow experience should be emphasized, and therefore, more immersive learning environments and content should be incorporated into the teaching process to ensure children's engagement in learning. Furthermore, the researchers provided recommendations for enhancing the depth of instructional content and for implementing grading systems that would foster competition and facilitate the establishment of learning objectives.

### 7.3. Limitations and future directions

The prototype for this study was developed based on the Western chivalric cultural background, which may lead to cultural cognitive bias in children from primary schools in China, thus affecting the research results. Due to the researchers' resource limitations, the sampling of this study was concentrated in two primary schools in one city, which resulted in the conclusions of this study not being able to be extrapolated to a regional or national scale. In addition, due to the young age of the student participants, their language expression ability is limited, and they are unable to have an in-depth understanding of virtual concepts such as "flow experience" and "learning engagement", which makes the qualitative results of this study relatively simple. The emphasis on children as the primary subjects of research has led to an insufficient incorporation of viewpoints from educators and parents in the resultant findings. Overall, the lack of breadth and diversity in this sampling may lead to a lack of representativeness in the research results.

In subsequent investigations, the researchers will focus on enhancing the functionality of the prototype. Once the prototype's features are sufficiently developed, the researchers will implement a pre-test/post-test experimental design

to obtain more comprehensive and precise test outcomes. Researchers may also consider surveying and interviewing educators and parents about their views on unplugged coding educational tools for children. In addition, researchers may explore the integration of gamification with educational tools across various disciplines, including art, language, and natural sciences, in order to evaluate the practical efficacy of gamified educational resources within these domains.

## Supporting information

**S1 File.** Appendices A and B.
(DOCX)

## Author contributions

**Conceptualization:** Lei Wang.

**Formal analysis:** Miao Huang.

**Methodology:** Miao Huang.

**Supervision:** Miao Huang.

**Visualization:** Lei Wang.

**Writing – original draft:** Miao Huang, Lei Wang.

**Writing – review & editing:** Lei Wang.

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
