## [Decision Letter · Decision Letter 0]

7 Feb 2025

PONE-D-24-57556Start Learning Coding without Computers? A Case Study on Children’s Gamified Coding Education Tool with Explanatory Sequential Mixed MethodPLOS ONE?

Dear Dr. Wang,

Thank you for submitting your manuscript to PLOS ONE. After careful consideration, we feel that it has merit but does not fully meet PLOS ONE’s publication criteria as it currently stands. Therefore, we invite you to submit a revised version of the manuscript that addresses the points raised during the review process.

We look forward to receiving your revised manuscript.

Kind regards,

Zhanni Luo

Academic Editor

PLOS ONE

Journal Requirements:

“This work is funded by the Jiangsu Provincial Educational Science Planning Project Key Project (2022), issued by the Jiangsu Provincial Education Science Planning Leading Group and Communication University of China, Nanjing. Project title: Innovative Research on Children’s Logic Teaching Practice in the Context of Gamification (B/2022/04/42).”

“This work is funded by the Jiangsu Provincial Educational Science Planning Project Key Project (2022), issued by the Jiangsu Provincial Education Science Planning Leading Group and Communication University of China, Nanjing. Project title: Innovative Research on Children’s Logic Teaching Practice in the Context of Gamification (B/2022/04/42)”

Additional Editor Comments (if provided):

1.I recommend organizing your discussion around specific themes.

2.The "design content" column in Table 1 needs to be more technical. The phrase "children will be rewarded for trying the games" is merely a noun definition of "reward". In the design section, you should explicitly outline the design logic and implementation criteria of the reward system, such as which types of rewards will be given for specific user behaviors and what those rewards are. Please revise accordingly.

3.In your hypotheses, you’ve prefixed each game element with the word “gamified”, such as “gamified challenges”, “gamified cooperation”, and “gamified role-playing”. I believe the term "gamified" should be removed.

4.The description of Table 5 is flawed. There are mistakes. Please revise.

5.You mention that “all variables have sufficient discriminant validity". Please provide the benchmark for clarity.

6.Please expand on the descriptions of Table 5 and Table 6.

7.The headers in the first and last columns of Table 7, “hypothesis/relationship” and “decision,” are inaccurate. These should be revised.

8.The header in Table 1 needs revision. The content in the first column should be summarized as something like “game elements”, not "gamification." Additionally, the wording of “design content” may not be appropriate. Please revise.

9.Table 2 needs headers. You need to include column headers so that we know how to interpret the data. For example, with a header of "frequency", we understand that 63 and 91 should be entered; with a header of "percentage", we know to enter 21.36% and 30.85%. What should the headers for “age” and “gender” be? Please add.

10.In Table 2, is it “coding experience” or “coding education experience”? The latter refers to the experience of teaching coding, right? Please revise.

11.Please avoid using symbols like “&” in the main text. The text should only contain English letters and Arabic numerals. For instance, it should be “Fornell and Larcker criterion”, not “Fornell & Larcker criterion.”

12.Please ensure consistency in the use of the term "Fornell-Larcker criterion". Currently, it appears interchangeably as "Fornell-Larcker criterion" and "Fornell and Larcker criterion".

13.When using abbreviations for the first time, provide their full form, e.g., S-O-R theory.

14.Please verify the use of single and double quotation marks to ensure they are applied correctly.

15.Please check whether capitalization is needed for any of the words in the table. Currently, you have expressions like “One Year and Above”.

16.Ensure consistency in the word forms for the elements in Table 10. For example, “to be hero” is a verb, while “teamwork” is a noun.

17.It’s recommended to adjust the font in Figure 1 to match the font used in the main text. Optional.

Reviewers' comments:

Reviewer's Responses to Questions

**Comments to the Author**

1. Is the manuscript technically sound, and do the data support the conclusions?

Reviewer #1: Yes

Reviewer #2: Partly

2. Has the statistical analysis been performed appropriately and rigorously?

Reviewer #1: Yes

Reviewer #2: Yes

3. Have the authors made all data underlying the findings in their manuscript fully available?

Reviewer #1: Yes

Reviewer #2: Yes

4. Is the manuscript presented in an intelligible fashion and written in standard English?

Reviewer #1: No

Reviewer #2: Yes

Reviewer #1: Overall Impression: The paper makes a valuable contribution to the field of gamified coding education, particularly in the context of unplugged tools for children. The use of a mixed-methods approach, robust statistical analysis, and integration of quantitative and qualitative results are notable strengths. However, the paper could benefit from a more detailed discussion of limitations and practical implications. With these improvements, the study would provide a more comprehensive and impactful contribution to the literature.

Strengths of the Paper

Robust Methodology:

The use of PLS-SEM for quantitative analysis and thematic analysis (TA) for qualitative data is appropriate and well-executed. The combination of these methods provides a comprehensive understanding of the research problem.

The inclusion of discriminant validity checks (Fornell & Larcker criterion and HTMT) ensures the reliability and validity of the measurement model.

Clear Presentation of Results:

The results are presented systematically, with tables and clear explanations of statistical findings (e.g., factor loadings, Cronbach’s alpha, AVE, and effect sizes).

The integration of quantitative and qualitative results using the pillar integration process (PIP) is a strong point, as it allows for a holistic interpretation of the data.

Theoretical Contributions:

The study successfully applies the S-O-R (Stimulus-Organism-Response) framework to explain the mediating role of flow experience in gamified coding education. This adds to the theoretical understanding of how gamification elements influence learning engagement.

The findings on the mediating effects of flow experience are well-supported and align with existing literature (e.g., Thomas & Baral, 2023).

Practical Implications:

The development and testing of the “Coding Adventure” prototype demonstrate the practical applicability of the research. The positive feedback from children and the successful field testing highlight the tool’s potential for real-world use.

The identification of effective gamification elements (e.g., role-playing, cooperation, and rewards) provides actionable insights for educators and developers of unplugged coding tools.

Discussion of Findings:

The discussion is thorough and well-structured, linking the results to existing literature and providing clear explanations for the observed effects (e.g., how role-playing and cooperation enhance flow experience and learning engagement).

Areas for Improvement or Clarification

Limitations of the Study:

While the limitations are mentioned (e.g., sampling from two schools in one city and children’s limited expressive ability), they could be expanded upon. For example:

How might the cultural context of China influence the generalizability of the findings?

Were there any challenges in implementing the unplugged coding tool in a classroom setting?

A more detailed discussion of these limitations would strengthen the paper’s transparency and credibility.

Qualitative Data Analysis:

The qualitative results section is relatively brief and lacks depth. While the researchers acknowledge the children’s limited ability to express abstract concepts like “flow experience,” more effort could be made to extract meaningful insights from the interviews.

Including direct quotes or anecdotes from the children could make the qualitative findings more engaging and relatable.

Effect Size Interpretation:

The interpretation of effect sizes (e.g., small, medium, large) is based on Kenny’s (2016) standards, but the paper does not discuss the practical significance of these effect sizes. For instance, what do these effect sizes mean in the context of children’s learning outcomes?

Mediating Effects:

While the mediating effects of flow experience are discussed, the explanation could be more detailed. For example:

How do the specific gamification elements (e.g., role-playing, cooperation) interact with flow experience to enhance learning engagement?

Are there any boundary conditions or moderating factors that influence these mediating effects?

Discussion of Rewards:

The discussion on gamified rewards is somewhat limited. While the paper mentions intrinsic and extrinsic rewards, it does not explore how these rewards might interact with other gamification elements (e.g., role-playing or cooperation) to influence learning engagement.

Future Research Directions:

The future directions section could be expanded to include more specific research questions or hypotheses. For example:

How might the effectiveness of gamification elements vary across different age groups or educational contexts?

What are the long-term effects of using unplugged gamified coding tools on children’s computational thinking skills?

Ethical Considerations:

The paper does not explicitly address ethical considerations, such as obtaining informed consent from parents or ensuring the well-being of child participants during the study. Including this information would enhance the paper’s rigor.

Suggestions for Improvement

Expand the Qualitative Analysis:

Provide more detailed insights from the interviews, including direct quotes or examples of children’s responses. This would make the qualitative findings more vivid and compelling.

Enhance the Discussion of Limitations:

Discuss the potential impact of cultural context, sampling limitations, and implementation challenges on the study’s findings. This would provide a more nuanced understanding of the study’s scope and generalizability.

Clarify Practical Implications:

Provide specific recommendations for educators and developers on how to implement the findings in real-world settings. For example, what are the key design principles for creating effective unplugged gamified coding tools?

Discuss Effect Sizes in Context:

Explain the practical significance of the effect sizes observed in the study. For instance, how might a “large” effect size translate into measurable improvements in children’s learning outcomes?

Explore Interactions Between Gamification Elements:

Investigate how different gamification elements (e.g., rewards, role-playing, cooperation) interact with each other to influence flow experience and learning engagement.

Reviewer #2: This paper explores gamified unplugged coding education for young learners, grounded in Stimulus-Organism-Response (S-O-R) theory. While the research topic is timely and relevant, the manuscript has several major weaknesses that must be addressed before it can be considered for publication. The theoretical contribution, though well-articulated, lacks sufficient critical engagement with prior work on gamification and coding education, especially in non-digital learning environments. Additionally, the research problem and hypotheses are formulated in a way that assumes gamification inherently improves learning outcomes rather than critically interrogating the conditions under which different elements succeed or fail. This assumption weakens the theoretical framing and should be revisited.

Methodologically, the study has some problems, from sampling limitations and potential biases that undermine the validity of its findings. The sample consists of only two schools in Nanjing, raising concerns about generalizability. There is no discussion of socio-economic or educational background factors, which are crucial in evaluating the effectiveness of gamified learning. Moreover, the PLS-SEM analysis lacks depth, particularly in its treatment of mediation effects and model robustness. The lack of a pre-test/post-test design makes it impossible to determine whether the observed effects are due to actual learning improvements or merely engagement during the gamified activity. The qualitative component is also underdeveloped—thematic analysis is presented descriptively rather than critically. The study would benefit from a more rigorous qualitative coding approach supported by inter-rater reliability checks.

A major issue is the superficial treatment of negative or neutral outcomes. The study claims that role-playing and cooperation significantly enhance the flow of experience and learning engagement, but it does not account for the weaker impact of challenges and rewards. Instead of reporting this finding, the authors should analyze why these elements underperformed—perhaps due to poor design, cognitive overload, or lack of individual motivation factors. Without a clear discussion of these inconsistencies, the conclusions feel overstated. Additionally, the long-term impact of the intervention is completely ignored. Gamification often produces short-term engagement spikes that do not translate into sustained learning benefits, a limitation that should have been explicitly addressed.

The practical implications of the study are also overstated. The "Coding Adventure" prototype is not rigorously tested against existing educational tools, making it difficult to determine whether its benefits are due to the gamification elements themselves or simply the novelty of the experience. There is no comparative analysis with digital coding tools or traditional instructional methods, which is a major oversight. Furthermore, the study ignores a crucial stakeholder group: parents and teachers. Given the emphasis on unplugged learning as an alternative to screen-based education, the study should include perspectives from educators and parents to evaluate whether such an approach is feasible for widespread adoption.

In conclusion, this paper requires major revisions before it can be considered for publication. The theoretical framing should be refined to critically engage with prior research, rather than assuming gamification’s benefits. The methodology must address its generalizability issues, incorporate a more rigorous qualitative approach, and critically assess underperforming gamification elements. Finally, the practical contributions need substantial revision, with comparative analysis and long-term evaluation. Until these concerns are addressed, the study remains an interesting but incomplete contribution to gamified coding education.

**Do you want your identity to be public for this peer review?** For information about this choice, including consent withdrawal, please see our Privacy Policy

Reviewer #1: **Yes: ** Samaa Mohammed Shohieb

Reviewer #2: No

---

## [Author Response · Author response to Decision Letter 1]

25 Feb 2025

Response to the editor’s comments and suggestions:

1 I recommend organizing your discussion around specific themes.

The discussion has been reorganized according to seven themes, corresponding to the above integration Table 10. (Section 6 / P-20)

2 The "design content" column in Table 1 needs to be more technical. The phrase "children will be rewarded for trying the games" is merely a noun definition of "reward". In the design section, you should explicitly outline the design logic and implementation criteria of the reward system, such as which types of rewards will be given for specific user behaviors and what those rewards are. Please revise accordingly.

Revised the columns of Table 1 and explained the reward design logic in detail. (Section 4.2 / P-10)

3 In your hypotheses, you’ve prefixed each game element with the word “gamified”, such as “gamified challenges”, “gamified cooperation”, and “gamified role-playing”. I believe the term "gamified" should be removed.

The word “gamified” was removed from all contents related to the hypotheses in the context. (Section 3 / P-8)

4 The description of Table 5 is flawed. There are mistakes. Please revise.

Revised. (Section 5.1.1 / P-13)

5 You mention that “all variables have sufficient discriminant validity". Please provide the benchmark for clarity.

All benchmarks related to discriminant validity have been re-stated. (Section 5.1.1 / P-13)

6 Please expand on the descriptions of Table 5 and Table 6.

Expanded. (Section 5.1.1 / P-13)

7 The headers in the first and last columns of Table 7, “hypothesis/relationship” and “decision,” are inaccurate. These should be revised.

Headers in Table 7-9 are revised. (Section 5.1.2 / P-14-15)

8 The header in Table 1 needs revision. The content in the first column should be summarized as something like “game elements”, not "gamification." Additionally, the wording of “design content” may not be appropriate. Please revise.

Revised. (Section 4.2 / P-10)

9 Table 2 needs headers. You need to include column headers so that we know how to interpret the data. For example, with a header of "frequency", we understand that 63 and 91 should be entered; with a header of "percentage", we know to enter 21.36% and 30.85%. What should the headers for “age” and “gender” be? Please add.

Added. (Section 4.3.1 / P-11)

10 In Table 2, is it “coding experience” or “coding education experience”? The latter refers to the experience of teaching coding, right? Please revise.

Revised. (Section 4.3.1 / P-11)

11 Please avoid using symbols like “&” in the main text. The text should only contain English letters and Arabic numerals. For instance, it should be “Fornell and Larcker criterion”, not “Fornell & Larcker criterion.”

“&” has been uniformly replaced by “and” in the context. (Section 5.1.1 / P-13)

12 Please ensure consistency in the use of the term "Fornell-Larcker criterion". Currently, it appears interchangeably as "Fornell-Larcker criterion" and "Fornell and Larcker criterion".

It has been determined to be the “Fornell-Larcker criterion.” (Section 5.1.1 / P-13)

13 When using abbreviations for the first time, provide their full form, e.g., S-O-R theory.

Revised in context. (Abstract / P-1)

14 Please verify the use of single and double quotation marks to ensure they are applied correctly.

Double quotes have been used throughout the text. (Full text)

15 Please check whether capitalization is needed for any of the words in the table. Currently, you have expressions like “One Year and Above”.

The capitalization of text in all tables has been unified. (Full text)

16 Ensure consistency in the word forms for the elements in Table 10. For example, “to be hero” is a verb, while “teamwork” is a noun.

The categories in Table 10 have been unified into noun forms. (Section 5.3 / P-16)

17 It’s recommended to adjust the font in Figure 1 to match the font used in the main text. Optional.

The font in Figures 1 and 2 has been unified with the font of the main text. (Section 3 / P-6; Section 4.1 / P-9)

Response to the reviewer#1’s comments and suggestions:

1 Expand the Qualitative Analysis: Provide more detailed insights from the interviews, including direct quotes or examples of children’s responses. This would make the qualitative findings more vivid and compelling.

Direct quotations from participants reinforced the credibility of findings and the effectiveness of explanations across all discussion themes. (Section 6 / P-20)

2 Enhance the Discussion of Limitations: Discuss the potential impact of cultural context, sampling limitations, and implementation challenges on the study’s findings. This would provide a more nuanced understanding of the study’s scope and generalizability.

The limitations section was strengthened by adding explanations of cultural differences and sampling diversity limitations. (Section 7.3 / P-22)

3 Clarify Practical Implications: Provide specific recommendations for educators and developers on how to implement the findings in real-world settings. For example, what are the key design principles for creating effective unplugged gamified coding tools?

The specific design experience and effective gamification elements gained from this study are supplemented to enhance practical significance. (Section 7.2 / P-21)

4 Discuss Effect Sizes in Context: Explain the practical significance of the effect sizes observed in the study. For instance, how might a “large” effect size translate into measurable improvements in children’s learning outcomes?

The effect size description and explanation has been incorporated into all themes in the discussion. (Section 6 / P-19)

5 Explore Interactions Between Gamification Elements: Investigate how different gamification elements (e.g., rewards, role-playing, cooperation) interact with each other to influence flow experience and learning engagement.

A paragraph was added to the discussion detailing the interrelationships of gamification elements and their impact on learning engagement. (Section 6 / P-20)

Response to the reviewer#2’s comments and suggestions:

1 This paper explores gamified unplugged coding education for young learners, grounded in Stimulus-Organism-Response (S-O-R) theory. While the research topic is timely and relevant, the manuscript has several major weaknesses that must be addressed before it can be considered for publication. The theoretical contribution, though well-articulated, lacks sufficient critical engagement with prior work on gamification and coding education, especially in non-digital learning environments. Additionally, the research problem and hypotheses are formulated in a way that assumes gamification inherently improves learning outcomes rather than critically interrogating the conditions under which different elements succeed or fail. This assumption weakens the theoretical framing and should be revisited.

Two paragraphs were added to the introduction to critically discuss the current status and problems of unplugged coding education and related research, in order to emphasize the necessity of this study. (Section 1 / P-3)

2 Methodologically, the study has some problems, from sampling limitations and potential biases that undermine the validity of its findings. The sample consists of only two schools in Nanjing, raising concerns about generalizability. There is no discussion of socio-economic or educational background factors, which are crucial in evaluating the effectiveness of gamified learning. Moreover, the PLS-SEM analysis lacks depth, particularly in its treatment of mediation effects and model robustness. The lack of a pre-test/post-test design makes it impossible to determine whether the observed effects are due to actual learning improvements or merely engagement during the gamified activity. The qualitative component is also underdeveloped—thematic analysis is presented descriptively rather than critically. The study would benefit from a more rigorous qualitative coding approach supported by inter-rater reliability checks.

The regret that a more comprehensive research design was not fully implemented in the methodology has been mentioned in Limitations and future directions, and it is hoped that they will be compensated in future research. (Section 7.3 / P-22)

3 A major issue is the superficial treatment of negative or neutral outcomes. The study claims that role-playing and cooperation significantly enhance the flow of experience and learning engagement, but it does not account for the weaker impact of challenges and rewards. Instead of reporting this finding, the authors should analyze why these elements underperformed—perhaps due to poor design, cognitive overload, or lack of individual motivation factors. Without a clear discussion of these inconsistencies, the conclusions feel overstated. Additionally, the long-term impact of the intervention is completely ignored. Gamification often produces short-term engagement spikes that do not translate into sustained learning benefits, a limitation that should have been explicitly addressed.

The specific design experience and effective gamification elements gained from this study are supplemented to enhance practical significance. (Section 7.2 / P-21)

4 The practical implications of the study are also overstated. The "Coding Adventure" prototype is not rigorously tested against existing educational tools, making it difficult to determine whether its benefits are due to the gamification elements themselves or simply the novelty of the experience. There is no comparative analysis with digital coding tools or traditional instructional methods, which is a major oversight. Furthermore, the study ignores a crucial stakeholder group: parents and teachers. Given the emphasis on unplugged learning as an alternative to screen-based education, the study should include perspectives from educators and parents to evaluate whether such an approach is feasible for widespread adoption.

The researchers included a comparison between the unplugged coding teaching tool and traditional teaching tools in the introduction, and criticized the shortcomings of the prototype design in the qualitative results and discussion sections. At the end, they also pointed out the lack of teacher and parent perspectives in the limitations. (Section 7.3 / P-22)

5 In conclusion, this paper requires major revisions before it can be considered for publication. The theoretical framing should be refined to critically engage with prior research, rather than assuming gamification’s benefits. The methodology must address its generalizability issues, incorporate a more rigorous qualitative approach, and critically assess underperforming gamification elements. Finally, the practical contributions need substantial revision, with comparative analysis and long-term evaluation. Until these concerns are addressed, the study remains an interesting but incomplete contribution to gamified coding education.

The researchers integrated all previous revision suggestions, focused on critically discussing the successful experiences and shortcomings of prototype development, revised the research methodology from a more comprehensive perspective, and strengthened the summary of significances and limitations. (Full text / Section 6 / P-20 )

---

## [Decision Letter · Decision Letter 1]

23 Apr 2025

PONE-D-24-57556R1Start Learning Coding without Computers? A Case Study on Children’s Gamified Coding Education Tool with Explanatory Sequential Mixed MethodPLOS ONE?

Dear Dr. Wang,

Thank you for submitting your manuscript to PLOS ONE. After careful consideration, we feel that it has merit but does not fully meet PLOS ONE’s publication criteria as it currently stands. Therefore, we invite you to submit a revised version of the manuscript that addresses the points raised during the review process.

We look forward to receiving your revised manuscript.

Kind regards,

Jin Su Jeong, Ph.D.

Academic Editor

PLOS ONE

Reviewers' comments:

Reviewer's Responses to Questions

**Comments to the Author**

Reviewer #1: All comments have been addressed

Reviewer #3: (No Response)

Reviewer #4: All comments have been addressed

2. Is the manuscript technically sound, and do the data support the conclusions?

Reviewer #1: Yes

Reviewer #3: Partly

Reviewer #4: Yes

3. Has the statistical analysis been performed appropriately and rigorously?

Reviewer #1: Yes

Reviewer #3: I Don't Know

Reviewer #4: Yes

4. Have the authors made all data underlying the findings in their manuscript fully available?

Reviewer #1: Yes

Reviewer #3: Yes

Reviewer #4: No

5. Is the manuscript presented in an intelligible fashion and written in standard English?

Reviewer #1: Yes

Reviewer #3: Yes

Reviewer #4: Yes

Reviewer #1: By combining gamification elements with hands-on experiences, your research has the potential to inspire educators and developers worldwide, offering a meaningful alternative to screen-based learning. Your effort to integrate mixed-methods research with theoretical depth showcases a meticulous and insightful approach that significantly contributes to the field of educational technology. This work is not only a valuable resource for advancing coding education for children but also a beacon of inspiration for future innovations. Thank you for your contributions to this essential domain of study. While your paper now is more reflective, there are some small modifications that can be performed

1-The analysis of weaker aspects, such as the "challenge" component, lacks depth. Additional reflection on cognitive load, design, or motivation might illuminate these inconsistencies.

2-Without comparative analysis against existing tools or teacher/parent perspectives, the claims of practical significance could be overstated. Including such comparisons would strengthen the argument.

3-The absence of a pre-test/post-test design restricts conclusions on actual learning improvements.

Reviewer #3: 1. In general, academic English should be significantly improved throughout the manuscript.

2. From my perspective, it would be interesting to explain in greater depth what this ‘unplugged gamified coding activity’(Coding Adventure) consists of and how it is developed in the classroom session. It would help the readers to better understand what it consists of, and it would be useful to be able to replicate future educational practices.

3. The title mentions 'coding without a computer', however it is not understood how this is carried out or the implications about it.

4. Gamification elements have been used, and it is intended to involve students in learning. However, it is important to identify what the learning objective is and what educational topics/content and/or competencies are intended to be worked on and acquired by the students with the application of this prototype and what educational results are to be achieved. Otherwise, it will lack an educational component and will be only a playful prototype.

5. In addition, the applied activity is sometimes referred to as “gamified coding activity” and sometimes as “unplugged gamified coding activity”. It is not very clear what the unplugged activity and the coding activity consist of, nor their implications in the results of this study, since what is analyzed are the implications of gamification elements.

6. Further information on hypothesis 9 is missing in section 3.

7. It is mentioned that the participants provided verbal informal consent. For future work, it would be interesting to include the bioethical code of the study, to inform the adults in charge of the underage students about the bioethics of the study to be able to participate in it and collect written information approval.

8. Justify text of tables 1,10 and 11.

9. In table 4, the acronyms of the constructions are already used, therefore in the next tables (7,9,10 and 11). As a suggestion, maybe it could be useful to use them to avoid large text within tables.

10. In Table 10, regarding ‘Pillar building themes’ maybe it could be more aesthetic to organize according to the order mentioned in the study theoretical framework: 1)RP, 2)R, 3)CH, 4)C 5)F and 6)LE.

11. The items from the quantitative questionnaire should be included. Also, the questions asked in the qualitative interview should also be specified.

12. The versions of the software used may be specified.

13. Regarding the analysis of the data, it might be useful to explain the section corresponding to Table 9 “Results of the tests of mediating effects (How was the analysis performed, and the results obtained? and explain the results). On the other hand, I do not recognize the hypotheses analyzed in these results in Table 9. What hypotheses do they correspond to in the theoretical framework created in Figure 1, since H5 relates only flow to learning engagement, not with specific elements of gamification. In table 9 three variables are analyzed while in the theoretical framework there is no hypothesis that relates to three variables. Is it possible to compare three variables with the T-student test?

14. It would be interesting to include more references in the different parts of the discussion.

15. Finally, the word document with further detailed revisions is attached.

Reviewer #4: Thank you for addressing the previous remarks. Please consider the following observations;

1) Section 3. Hypothesis 9 doesn't include a description of the importance of such a hypothesis

2) Section 4.1. This section is written in the future tense, but this study has already taken place, please rewrite this section using the correct tense.

3) Section 4.2. We don't know characteristics of the students that participated in the study, demographics, besides gender and age? It is hard then to find practical implications for those that might share similar background or circumstances.

4) Section 4.3.1 There is no example of the type of questions that were used in the study, no appendix or URL where someone could check the questionnaire. So this is missing.

5) Section 4.3.2 No sample of the questions used in the interviews. Again, no external URL or appendix to read the questions you applied in the interviews,

6) Section 5.1.2 F2 for effect size is related mostly as Cohen effect size. Is it possible to use this more standard way to refer to effect sizes?

7) Section 5.2 When the reviewer was reading this section there was no "warning" that indicated the data for the inteview was going to be presented in section 5.3... So there is a missing paragraph that should explain how the interview data was going to be treated.

8) Section 7.3 does not covers that there is missing information about the background of the students participating in the study. Please review this

**Do you want your identity to be public for this peer review?** For information about this choice, including consent withdrawal, please see our Privacy Policy

Reviewer #1: No

Reviewer #3: No

Reviewer #4: No

---

## [Author Response · Author response to Decision Letter 2]

22 May 2025

Reviewer #1:

1. The analysis of weaker aspects, such as the "challenge" component, lacks depth. Additional reflection on cognitive load, design, or motivation might illuminate these inconsistencies.

Revision: The possible reasons for the weak effect of the "challenge" element in the prototype of this study are further analyzed in the penultimate paragraph of the discussion.

2. Without comparative analysis against existing tools or teacher/parent perspectives, the claims of practical significance could be overstated. Including such comparisons would strengthen the argument.

Revision: Since the research design did not include feedback from teachers and parents, it is not possible to include relevant discussion in the analysis of the results, but this has been added as limitations in Section 7.3 at the end of the paper. Suggestions are also made in the future research directions.

3. The absence of a pre-test/post-test design restricts conclusions on actual learning improvements.

Revision: In the last paragraph of Section 7.3, the researchers added a revision mentioning this limitation and considering it as a future research direction.

Reviewer #3:

1. In general, academic English should be significantly improved throughout the manuscript.

Revision: The parts of the paper that were not written in academic English have been polished.

2. From my perspective, it would be interesting to explain in greater depth what this ‘unplugged gamified coding activity’(Coding Adventure) consists of and how it is developed in the classroom session. It would help the readers to better understand what it consists of, and it would be useful to be able to replicate future educational practices.

Revision: We have added a whole paragraph to section 4.2 describing the game rules and how to build the prototype and added test photos for visual demonstration.

3. The title mentions 'coding without a computer', however it is not understood how this is carried out or the implications about it.

Revision: The main purpose of this article is to show that it is possible to learn coding without a computer, and it is the beginning stage of learning, which is already indicated in the title. In addition, the researchers also further emphasized the possibility of teaching in a real environment without computers in the prototype design description in Section 4.2.

4. Gamification elements have been used, and it is intended to involve students in learning. However, it is important to identify what the learning objective is and what educational topics/content and/or competencies are intended to be worked on and acquired by the students with the application of this prototype and what educational results are to be achieved. Otherwise, it will lack an educational component and will be only a playful prototype.

Revisions: The following teaching concept has been more clearly explained in Section 4.2. According to the common planning of coding education, the researchers designed the introductory coding statements as command cards in the prototype. By repeatedly using command cards in the game, learners are familiar with the basic logic of coding and reasonably plan the number of times these statements are used. With more challenging puzzle tasks brought by level upgrades, learners gradually understand the use of complex coding statements and master the thinking skills to solve complex problems. The coding knowledge acquired through the above training has become the basis for learners to continue learning CT.

5. In addition, the applied activity is sometimes referred to as “gamified coding activity” and sometimes as “unplugged gamified coding activity”. It is not very clear what the unplugged activity and the coding activity consist of, nor their implications in the results of this study, since what is analyzed are the implications of gamification elements.

Revisions: Firstly, the researchers clearly stated in the title of the paper that the research focus is "unplugged" and included it in the title. Secondly, in the penultimate paragraph of the first chapter, regarding the teaching effect of unplugged coding education, the original point of view focused on the gaming effect. After the revision, more discussions from the teaching effect perspective were added.

6. Further information on hypothesis 9 is missing in section 3.

Revisions: The information in hypothesis 9 in Section 3 has been strengthened. Also, because the format of the paper is to cite previous studies first and then summarize and propose hypotheses, the reviewer may not have noticed the citation paragraph before the hypothesis in the original version.

7. It is mentioned that the participants provided verbal informal consent. For future work, it would be interesting to include the bioethical code of the study, to inform the adults in charge of the underage students about the bioethics of the study to be able to participate in it and collect written information approval.

Revisions: Thank you for your suggestion. In fact, follow-up research is already underway.

8. Justify text of tables 1,10 and 11.

Revisions: Already adjusted.

9. In table 4, the acronyms of the constructions are already used, therefore in the next tables (7,9,10 and 11). As a suggestion, maybe it could be useful to use them to avoid large text within tables.

Revisions: They have all been adjusted to ensure that abbreviations are used to improve reading efficiency without affecting comprehension.

10. In Table 10, regarding ‘Pillar building themes’ maybe it could be more aesthetic to organize according to the order mentioned in the study theoretical framework: 1)RP, 2)R, 3)CH, 4)C 5)F and 6)LE.

Revisions: Considering that Table 10 is based on the order of hypothesis numbers, and the order of the hypotheses is arranged according to the order of influence of the SOR theory, the current order is retained. However, the authors added hypothesis numbers to clarify the order and avoid misunderstanding by readers.

11. The items from the quantitative questionnaire should be included. Also, the questions asked in the qualitative interview should also be specified.

Revisions: All the items and questions included in the questionnaire and interview have been included in Appendices A and B.

12. The versions of the software used may be specified.

Revisions: The version number of the analysis software has been added at the beginning of Chapter 5.

13. Regarding the analysis of the data, it might be useful to explain the section corresponding to Table 9 “Results of the tests of mediating effects (How was the analysis performed, and the results obtained? and explain the results). On the other hand, I do not recognize the hypotheses analyzed in these results in Table 9. What hypotheses do they correspond to in the theoretical framework created in Figure 1, since H5 relates only flow to learning engagement, not with specific elements of gamification. In table 9 three variables are analyzed while in the theoretical framework there is no hypothesis that relates to three variables. Is it possible to compare three variables with the T-student test?

Revisions: The measurement of the mediating effect is indeed an extended part of the study, and no hypotheses were proposed before. Considering that the measurement results of this part did not affect the main conclusions of this study, it is considered to be deleted in this version of the revisions to avoid more space affecting the main part of the study and to ensure the rigor of the hypotheses and verifications.

14. It would be interesting to include more references in the different parts of the discussion.

Revisions: More literature citations have been added to the discussion to increase the richness of the discussion.

Reviewer #4:

1. Section 3. Hypothesis 9 doesn't include a description of the importance of such a hypothesis

Revisions: The information in hypothesis 9 in Section 3 has been strengthened. Also, because the format of the paper is to cite previous studies first and then summarize and propose hypotheses, the reviewer may not have noticed the citation paragraph before the hypothesis in the original version.

2. Section 4.1. This section is written in the future tense, but this study has already taken place, please rewrite this section using the correct tense.

Revisions: Section 4.1 has been mostly revised to ensure the use of correct tense and more academic language.

3. Section 4.2. We don't know characteristics of the students that participated in the study, demographics, besides gender and age? It is hard then to find practical implications for those that might share similar background or circumstances.

Revisions: Demographic data for quantitative studies are in 4.3.1, and demographic data for qualitative studies are in 4.3.2.

4. Section 4.3.1 There is no example of the type of questions that were used in the study, no appendix or URL where someone could check the questionnaire. So this is missing.

Revisions: The questionnaire for the quantitative study has been included in Appendix A.

5. Section 4.3.2 No sample of the questions used in the interviews. Again, no external URL or appendix to read the questions you applied in the interviews

Revisions: The interview questions for the qualitative study have been included in Appendix B.

6. Section 5.1.2 F2 for effect size is related mostly as Cohen effect size. Is it possible to use this more standard way to refer to effect sizes?

Revisions: After discussion, the researchers decided to keep the original reference. On the one hand, the standard cited by the researchers is more updated, and on the other hand, it is more in line with the situation of this study.

7. Section 5.2 When the reviewer was reading this section there was no "warning" that indicated the data for the inteview was going to be presented in section 5.3... So there is a missing paragraph that should explain how the interview data was going to be treated.

Revisions: Instructions for coding the data for the qualitative study have been added to Section 5.2, which also presents the coding results and a preview of the final integration with the quantitative data.

8. Section 7.3 does not covers that there is missing information about the background of the students participating in the study. Please review this

Revisions: The student participants’ age, gender, grade level, and coding learning experience are described in quantitative and qualitative demographic tables (Tables 2 and 3).

---

## [Decision Letter · Decision Letter 2]

6 Jun 2025

PONE-D-24-57556R2Start Learning Coding without Computers? A Case Study on Children's Unplugged Gamified Coding Education Tool with Explanatory Sequential Mixed MethodPLOS ONE?

Dear Dr. Wang,

Thank you for submitting your manuscript to PLOS ONE. After careful consideration, we feel that it has merit but does not fully meet PLOS ONE’s publication criteria as it currently stands. Therefore, we invite you to submit a revised version of the manuscript that addresses the points raised during the review process.

We look forward to receiving your revised manuscript.

Kind regards,

Jin Su Jeong, Ph.D.

Academic Editor

PLOS ONE

Journal Requirements:

Reviewers' comments:

Reviewer's Responses to Questions

**Comments to the Author**

Reviewer #1: All comments have been addressed

Reviewer #3: All comments have been addressed

2. Is the manuscript technically sound, and do the data support the conclusions?

Reviewer #1: Yes

Reviewer #3: Yes

3. Has the statistical analysis been performed appropriately and rigorously?

Reviewer #1: Yes

Reviewer #3: Yes

4. Have the authors made all data underlying the findings in their manuscript fully available?

Reviewer #1: Yes

Reviewer #3: Yes

5. Is the manuscript presented in an intelligible fashion and written in standard English?

Reviewer #1: Yes

Reviewer #3: Yes

Reviewer #1: The paper presents a well-structured and insightful discussion on the topic, providing thorough analysis, relevant supporting evidence, and clear conclusions. The research methodology is sound, ethical considerations are appropriately addressed, and the findings contribute meaningfully to the field. No concerns regarding dual publication or research ethics have been identified. The manuscript is well-prepared for acceptance, and I recommend its publication without further revision.

!

Reviewer #3: 1. Revise the citations, bibliographic references, tables and figures thet should follow the style specified by the journal.

2. All the comments to be adressed have been added in the word document attached.

**Do you want your identity to be public for this peer review?** For information about this choice, including consent withdrawal, please see our Privacy Policy

Reviewer #1: **Yes: ** Samaa Shohieb

Reviewer #3: No

---

## [Author Response · Author response to Decision Letter 3]

17 Jun 2025

Dear reviewer:

Since the revisions are all for the details of the sentences and formats in the manuscript, they are all marked in the manuscript. Please enter the manuscript for review.

Thank you.

---

## [Decision Letter · Decision Letter 3]

30 Jun 2025

PONE-D-24-57556R3Start Learning Coding without Computers? A Case Study on Children's Unplugged Gamified Coding Education Tool with Explanatory Sequential Mixed MethodPLOS ONE?

Dear Dr. Wang,

Thank you for submitting your manuscript to PLOS ONE. After careful consideration, we feel that it has merit but does not fully meet PLOS ONE’s publication criteria as it currently stands. Therefore, we invite you to submit a revised version of the manuscript that addresses the points raised during the review process.

We look forward to receiving your revised manuscript.

Kind regards,

Jin Su Jeong, Ph.D.

Academic Editor

PLOS ONE

Journal Requirements:

Reviewers' comments:

Reviewer's Responses to Questions

**Comments to the Author**

Reviewer #3: All comments have been addressed

2. Is the manuscript technically sound, and do the data support the conclusions?

Reviewer #3: Yes

3. Has the statistical analysis been performed appropriately and rigorously?

Reviewer #3: Yes

4. Have the authors made all data underlying the findings in their manuscript fully available?

Reviewer #3: Yes

5. Is the manuscript presented in an intelligible fashion and written in standard English?

Reviewer #3: Yes

Reviewer #3: Some comments have been added within the Word document attached, it can be highlighted the following revisions:

1.Recommendations of the paragraph before the hypotesis: to mention the four gamification elements

2.Add a paragraph explaining hypostheis 9

3.Formal and academic writting

4.Coherence thoroughout the whole document

**Do you want your identity to be public for this peer review?** For information about this choice, including consent withdrawal, please see our Privacy Policy

Reviewer #3: No

---

## [Author Response · Author response to Decision Letter 4]

5 Jul 2025

Dear reviewer:

1. Recommendations of the paragraph before the hypothesis: to mention the four gamification elements

Reply: Revised

2. Add a paragraph explaining hypothesis 9

Reply: In the previous revision, we tried to add an extended paragraph to hypothesis 9, but further revision opinions suggested deleting it to keep it on par with other hypothesis development paragraphs. In addition, the authors have discussed and felt that hypothesis 6-9 are of the same level and have similar content, so the current paragraph layout is relatively even. Therefore, we sincerely suggest that the editor consider maintaining the status quo.

3. Formal and academic writing

Reply: Revised

4. Coherence throughout the whole document

Reply: Made the following revision----

Corrected some minor formatting issues in the manuscript.

Summarized the comprehensive impact of gamification elements on flow experiences before H1-4 and added an overall description.

Adjusted the paragraph layout of 4.1 research design.

Added a complete description of the five-point Likert scale.

---

## [Decision Letter · Decision Letter 4]

8 Aug 2025

Start Learning Coding without Computers? A Case Study on Children's Unplugged Gamified Coding Education Tool with Explanatory Sequential Mixed Method

PONE-D-24-57556R4

Dear Dr. Wang,

We’re pleased to inform you that your manuscript has been judged scientifically suitable for publication and will be formally accepted for publication once it meets all outstanding technical requirements.

Kind regards,

Tomislav Jagušt

Academic Editor

PLOS ONE

Additional Editor Comments (optional):

Dear authors of manuscript "Start Learning Coding without Computers? A Case Study on Children's Unplugged Gamified Coding Education Tool with Explanatory Sequential Mixed Method",

I am pleased to inform you that the peer review process for your manuscript has been successfully completed, and the editorial decision is: Accept.

Thank you for choosing PLOS ONE for publishing your research results. We appreciate your contribution to the scientific community and hope you will consider submitting your future research to PLOS One as well. I also wish you a lot of success in your future academic and research career.

Best regards,

Tomislav Jagušt

Reviewers' comments:

Reviewer's Responses to Questions

**Comments to the Author**

Reviewer #3: All comments have been addressed

Reviewer #5: All comments have been addressed

2. Is the manuscript technically sound, and do the data support the conclusions?

Reviewer #3: Yes

Reviewer #5: Yes

3. Has the statistical analysis been performed appropriately and rigorously?

Reviewer #3: Yes

Reviewer #5: Yes

4. Have the authors made all data underlying the findings in their manuscript fully available?

Reviewer #3: Yes

Reviewer #5: Yes

5. Is the manuscript presented in an intelligible fashion and written in standard English?

Reviewer #3: Yes

Reviewer #5: Yes

Reviewer #3: No comments to the authors

Reviewer #5: The manuscript is well written and clearly structured. The topic is timely and relevant, addressing an important research area.

The theoretical framing using the S-O-R model is appropriate, and the application of an explanatory sequential mixed methods design is well justified. The quantitative and qualitative findings are thoroughly integrated, offering both statistical robustness and practical insight. The prototype "Coding Adventure" is thoughtfully designed, and the results are clearly presented and discussed in light of existing literature.

Overall, the manuscript makes a valuable contribution to the field of educational technology and early coding education.

**Do you want your identity to be public for this peer review?** For information about this choice, including consent withdrawal, please see our Privacy Policy

Reviewer #3: No

Reviewer #5: No

---

## [Editor Report · Acceptance letter]

PONE-D-24-57556R4

PLOS ONE

Dear Dr. Wang,

I'm pleased to inform you that your manuscript has been deemed suitable for publication in PLOS ONE. Congratulations! Your manuscript is now being handed over to our production team.

Kind regards,

on behalf of

Dr. Tomislav Jagušt

Academic Editor

PLOS ONE